# Salicylic Acid Regulates Root Gravitropic Growth via Clathrin-Independent Endocytic Trafficking of PIN2 Auxin Transporter in *Arabidopsis thaliana*

**DOI:** 10.3390/ijms23169379

**Published:** 2022-08-19

**Authors:** Houjun Zhou, Haiman Ge, Jiahong Chen, Xueqin Li, Lei Yang, Hongxia Zhang, Yuan Wang

**Affiliations:** 1The Engineering Research Institute of Agriculture and Forestry, Ludong University, Yantai 264025, China; 2College of Life Sciences, Nanjing University, Nanjing 210093, China; 3CAS Center for Excellence in Molecular Plant Sciences, Shanghai 201602, China

**Keywords:** *Arabidopsis thaliana*, auxin, gravitropism, PIN2, salicylic acid

## Abstract

The phytohormone salicylic acid (SA) plays a crucial role in plant growth and development. However, the mechanism of high-concentration SA-affected gravitropic response in plant root growth and root hair development is still largely unclear. In this study, wild-type, *pin2* mutant and various transgenic fluorescence marker lines of *Arabidopsis thaliana* were investigated to understand how root growth is affected by high SA treatment under gravitropic stress conditions. We found that exogenous SA application inhibited gravitropic root growth and root hair development in a dose-dependent manner. Further analyses using DIRECT REPEAT5 (DR5)-GFP, auxin sensor DII-VENUS, auxin efflux transporter PIN2-GFP, trans-Golgi network/early endosome (TGN/EE) clathrin-light-chain 2 (CLC2)-mCherry and prevacuolar compartment (PVC) (Rha1)-mCherry transgenic marker lines demonstrated that high SA treatment severely affected auxin accumulation, root-specific PIN2 distribution and *PIN2* gene transcription and promoted the vacuolar degradation of PIN2, possibly independent of clathrin-mediated endocytic protein trafficking. Our findings proposed a new underlying mechanism of SA-affected gravitropic root growth and root hair development via the regulation of *PIN2* gene transcription and PIN2 protein endocytosis in plants.

## 1. Introduction

Plant growth and response to environmental alterations are precisely governed by phytohormones. The phytohormone salicylic acid (SA), which functions as a plant defense activator, plays a crucial role in the local and systemic response against microbial pathogens and in the defining of the transduction pathway mediating plant response to abiotic stresses such as drought [1,2,3], salt stress [4], chilling [5,6], heavy metal tolerance [7,8,9] and heat [10,11].

In addition to its regulatory role in plant response to biotic and abiotic stresses, SA also has specific effects on plant growth and development [1]. It regulates various biological processes such as seed germination, bud development, vegetative growth, photosynthesis, respiration, thermogenesis, flower formation, fruit setting and ripening, seed production, senescence and cell death [1,12]. These effects are SA concentration-, plant growth condition- and developmental stage-dependent. Generally, a low concentration of SA promotes plant growth under unfavorable conditions, whereas a high concentration of SA inhibits plant growth [1,12,13]. In *Arabidopsis*, exogenous application of SA proportionally reduced root elongation [13,14,15] and specifically induced its waving growth in a concentration-dependent manner [15,16]. At the concentration of 250 µM, exogenous application of SA inhibited the growth of primary roots and the development of lateral roots [17]. In other plant species, such as maize, soybean and pine tree, a low concentration of SA increased root biomass [13,16,18,19,20]. More detailed studies demonstrated that SA regulated root development by affecting auxin signaling and auxin transport [13,15,16,17,21]. The number of periclinal and tangential divisions in the outer layers of roots was increased by SA-mediated auxin accumulation via a CYCD6;1-dependent mechanism [16]. Thus, a low concentration of SA promoted adventitious root development and altered root apical meristem architecture, whereas a high concentration of SA inhibited all the processes of root growth [16].

Since the discovery of directional auxin transport upon gravitropic stimulation, its mechanism has been well established in plants [22,23]. Gravitropic stimulation induced the asymmetric movement of auxin, which in turn caused the gravitropic curvature of root growth [24,25,26]. Therefore, auxin was essential for gravitropism [27]. In *Arabidopsis*, the root-specific PIN-FORMED2 (PIN2), which is shoot-ward localized in the lateral root cap and root epidermis cells, and root-ward localized in the root cortex cells, functions as an auxin efflux transporter directing the transport of auxin from root tip into the root elongation zone. During the gravitropic response, PIN2 regulated the transport of auxin [28]. In the *Arabidopsis pin2* mutant, shoot-ward auxin distribution in the lower side of roots was largely repressed during gravity stimulus, leading to the agravitropic growth of roots [29]. Meanwhile, PIN2 can also affect the onset and growth of root hairs [30]. Root hair-specific overexpression of PIN2 significantly inhibited root hair growth by depleting the auxin levels in root hair cells [31].

Recently, a negative correlation between SA and gravitropism was reported [32]. SA treatment resulted in a noticeable reduction in root gravitropism. Auxin redistribution and quantitative analyses suggested that SA reduced the lateral diffusion and endocytic internalization of PIN2 [32]. In this study, we investigated the role of SA response in root growth and root hair development upon gravity stimulus. We showed that exogenous application of high concentration of SA affected root development in a concentration-dependent manner upon gravity stimulus. We also demonstrated that the application of a high concentration of SA not only repressed the transcriptional expression of *PIN2* in the nucleus but also modulated the vacuolar degradation, possibly via the clathrin-independent endocytic trafficking, of PIN2 protein in *Arabidopsis*.

## 2. Results

### 2.1. High-Concentration SA Application Affects Root Gravitropic Growth in a Dose-Dependent Manner 

To understand whether exogenous application of high-concentration SA would affect the root growth and development of *Arabidopsis* in response to gravitropic stress, five-day-old wild-type (WT) *Arabidopsis* seedlings germinated on half-strength Murashige and Skoog (½MS) medium were subjected to different concentrations of SA treatments. *Arabidopsis* seedlings were transferred onto ½MS medium plates supplemented with 0, 100, 200 and 300 μM SA and cultured for 16 h after being orientated at 90° (Figure 1A). The deviated root tip angles along the vertical direction were measured. Consistent with previous reports, a dose-dependent agravitropic response in the root tip growth was observed [15,16,21,32,33,34]. At all concentrations tested, exogenous application of SA significantly reduced the deviated root tip angles and the length of newly grown roots (Figure 1A–C). We further examined the effects of SA treatments on root hair development and found that high-concentration SA treatments also significantly reduced root hair number and length in an SA-dose-dependent manner (Figure 1D–F). Compared to the control seedlings, root hair development of seedlings treated with a high concentration of SA was severely suppressed.

### 2.2. SA Reduces Auxin Accumulation and Distribution in the Root Apical Meristem

Auxin polar transport is the driving factor that affects root gravitropism [29]. We analyzed auxin distribution and polar transportation in the roots of *Arabidopsis* seedlings treated with 0 or 200 μM SA using the auxin response reporter DIRECT REPEAT5 (DR5)-GFP [35] and auxin sensor DII-VENUS [36] transgenic marker lines. We observed that DR5-dependent GFP fluorescence was mainly distributed in the root apical meristem (RAM) around the quiescent centre (QC) and columella cell (CC) areas. SA treatment significantly decreased the DR5-dependent GFP fluorescence in the QC and CC regions, whereas the DII-VENUS signal was gradually enhanced (Figure 2A–D). We further examined auxin accumulation after 90° re-orientation in the transgenic DR5-GFP seedlings. Without SA treatment, auxin was accumulated in the lower side of the root meristematic zone region upon gravity stimulus (Figure 2E). Consistent with a previous study, SA treatment decreased auxin accumulation in the lower side [32]. Previous time-course studies have shown that exogenous SA application suppressed auxin transport in the roots [16,32,34]. We also observed that after being exposed to 200 µM SA for 24 h, DR5-dependent GFP fluorescence was significantly reduced, and there was no auxin accumulation in the lower side of the roots (Figure 2E,F).

### 2.3. SA Reduces PIN2 Accumulation and Distribution in the Roots

As an auxin efflux transporter, root-specific PIN2 affected both the onset and growth of root hairs [30]. To understand whether SA treatment would affect the accumulation of PIN2 in roots, we investigated the protein dynamics of PIN2 in transgenic ProPIN2:*PIN2-GFP* seedlings [37]. Five-day-old ProPIN2:*PIN2-GFP* seedlings were transferred to ½MS medium containing 0 or 200 μM SA and cultured for 2 days. We found that a high dosage of SA treatments significantly decreased the accumulation of PIN2-GFP in roots, leading to a discontinuous PIN2-GFP signal on the plasma membrane (PM) (Figure 3A,B). We further analyzed *PIN2* gene expression in wild-type *Arabidopsis* and root growth in *PIN2* loss-of-function mutant (*pin2*) in response to high-concentration SA treatments. Seven-day-old *Arabidopsis* (Col-0) seedlings were treated with 200 μM SA for 0 h, 6 h, 12 h and 24 h, respectively. qRT-PCR analyses showed that *PIN2* expression was gradually suppressed for the duration of SA treatment (Figure 3C). Consistently, the root growth of *pin2-2* mutant seedlings showed an un-sensitive response to SA treatment. Unlike the wild-type seedlings, whose root growth on ½MS medium containing 20 μM SA was severely suppressed, the root growth of *pin2-2* seedlings was not affected (Appendix A). Therefore, high-concentration SA application might inhibit the seedling root growth by affecting both PIN2 expression and accumulation.

### 2.4. SA-Induced PIN2 Internalization Is Independent of Clathrin-Mediated Endocytosis

To further understand how PIN2 is impacted by a high concentration of SA application, we investigated its protein dynamics on the cell surface in the five-day-old ProPIN2:*PIN2-GFP* transgenic marker line seedlings grown on ½MS medium supplemented with 200 μM SA for 2, 3, 4, 6 and 24 h, respectively [37]. We found that, consistent with a recent report [32], the PIN2-GFP signal in the roots of seedlings treated with 200 μM SA was heterogeneous in shape and size and discontinuously displayed on the plasma membrane (PM) (Figure 4). Further quantitative analysis of the PIN2-GFP signal revealed that a high concentration of SA treatment decreased the incidence of PIN2-GFP at the PM in a time-dependent manner (Figure 4). In addition, the PIN2-GFP signal in the cytoplasm was observed in a dispersive state after being treated with 200 μM SA for 6 h, suggesting that high-concentration SA might have induced the entry of PIN2 into the vacuole for degradation.

Based on the above observations, we postulated that high-dosage SA treatment might promote PIN2 endocytosis. To verify this hypothesis, we separately crossed the PIN2-GFP transgenic line with the transgenic marker lines stably expressing the fluorescent markers for trans-Golgi network/early endosome (TGN/EE) clathrin-light-chain 2 (CLC2)-mCherry (CS781677) [38] and prevacuolar compartment (PVC) (Rha1-mCherry, CS781672) [39]. Five-day-old seedlings harboring both PIN2-GFP and CLC2-mCherry or PIN2-GFP and Rha1-mCherry were, respectively, treated with 200 μM SA for 2, 3, 5 and 16 h. The co-localizations of PIN2-GFP with these marker proteins in root cells were analyzed. We found that high-level SA application enhanced the intracellular accumulation of PIN2 in the punctate structures after 2 h, 3 h and 5 h treatments. Interestingly, the SA-induced clusters of PIN2-GFP and CLC2-mCherry did not exhibit apparent co-localization (Figure 5A). By contrast, PIN2-GFP exhibited obvious co-localization with Rha1-mCherry (Figure 5B). After 16 h SA treatment, PIN2-GFP concentrated in the structures that resembled vacuoles, suggesting that PIN2 could have been transported to the vacuole for subsequent degradation (Figure 5B).

## 3. Discussion

SA regulated not only the resistance to biotic and abiotic stresses but also the growth and development of plants [1,14,16]. Similar to strigolactones, which interact with nitric oxide and the effects of auxin on PIN2 targeting, trafficking and clathrin-mediated endocytosis to regulate root system architecture in *Arabidopsis thaliana* and pea [40], and function as signal molecules for the communication between plants and bacteria, SA also regulates plant root development by altering auxin signaling and transport [13,15,16,17,21,41]. Recently, a regulatory mechanism of PIN2-dependent auxin transport in response to a high-concentration SA treatment was reported [32]. Since PIN2 also affects the onset and growth of root hairs, we investigated the role of SA on both root growth and root hair development upon gravity stimulus. Statistical analysis of the root growth of *Arabidopsis* seedlings exposed to high concentrations of SA upon gravity stimulus exhibited significant inhibition of root growth and root hair development (Figure 1A–F). To understand how the root growth was affected by SA application, we treated the seedlings of the DR5-GFP marker line with 200 μM SA and examined the distribution of the DR5-GFP signal. As indicated by the decreased fluorescence signal in the *Arabidopsis* roots, a high concentration of SA treatment reduced auxin accumulation and distribution in the root apical meristem (Figure 2A–E).

During the gravitropic response, auxin transporter PIN2 was shown to be expressed in the cell layers of the epidermis and cortex to regulate auxin transport [28]. Under 100 µM of SA treatment, gravity triggered the redistribution of PIN2 [32]. Consistently, we observed that the application of 200 µM SA also led to impaired PIN2 accumulation and distribution (Figure 3A,B). SA can converge with jasmonic acid (JA), ethylene (ET), gibberellin (GA), abscisic acid (ABA) and auxin signals to affect plant growth and development by regulating gene transcription in the nucleus [12,21,42]. We observed that SA treatment down-regulated the transcription of *PIN2* in Col-0 seedlings (Figure 3C). Root hair-specific overexpression of *PIN2* greatly inhibited root hair growth by depleting auxin levels in the root hair cells [31]. We concluded that by regulating the expression of *PIN2* and the degradation of PIN2 protein, high levels of SA impaired PIN2-dependent auxin transport and inhibited root hair growth. Differently, exogenous ABA application reduced the intensity of membrane PIN2 by suppressing *PIN2* expression rather than by accelerating PIN2 degradation to elicit the waving root growth trajectory [43]. At a concentration of 100 μM, SA stimulated PIN2 hyperclustering during gravitropism [32]. We treated *Arabidopsis* seedlings with 200 μM SA and found that SA induced clathrin-independent mediated PIN2 degradation, suggesting different effects of different SA dosages on PIN2 (Figure 4).

A growing number of studies have demonstrated the importance of endocytosis in different physiological processes in plants [21]. Endocytosis that depends on the vesicle coat protein clathrin or clathrin-mediated endocytosis (CME) is the most prominent endocytic mechanism [21]. Cargo proteins can be endocytosed by the clathrin-dependent mechanism, which is constitutively or ligand inducibly dependent on the large GTPase dynamin, or by the clathrin-independent (CI) mechanism, which is either caveolarly and RhoA-regulatorily dependent on the large GTPase dynamin, or CDC42- and ARF6-regulatorily independent on the large GTPase dynamin [44]. CME not only functions in plant immune responses but also plays an essential role in nutrient uptake and intercellular transport of auxin, specifically in the internalization of the PIN family [21]. Auxin has been reported to inhibit the internalization step of PIN2 [13]. Brassinosteroid (BR) signaling acted as an antagonist of PIN2 endocytosis, thereby delimiting root gravitropism [45]. Now it is generally accepted that there are different clathrin-independent mechanisms, some of them regulated by ligands and membrane lipid composition [45,46]. In *Arabidopsis*, ABA can employ asparagine rich protein (NRP)-dependent PIN2 vacuolar degradation to suppress auxin-mediated primary root elongation [47]. In addition, high-level H_2_O_2_ can affect actin dynamics and thus modulate ARF-GEF-dependent trafficking of PIN2 [48]. Recently, protein phosphatase 2A (PP2A) was found to be a target of SA. SA binds to the subunit of PP2A to repress its dephosphorylation activity toward PIN proteins, leading to the hyperphosphorylation of PIN and a decrease in PIN activity, which subsequently decreased auxin export and attenuated plant growth [34]. Further study has shown that the M3 phosphorylation site was required for the trafficking and biological roles of PIN1, 2 and 7 [49]. Therefore, PP2A possibly affects the phosphorylation state of PIN2 and then regulates its trafficking. During pathogen-host interaction, SA showed inhibitory effects on PIN endocytosis and influenced PIN-modulated root architecture in a concentration-dependent manner [15,16,21,32]. A recent study also showed that SA induced membrane nanodomain compartmentalization and then suppressed clathrin-mediated PIN2 endocytosis [32]. We found that 200 µM SA-induced clusters of PIN2 were not co-localized with CLC2 but apparently co-localized with Rha1, suggesting that SA-induced PIN2 endocytosis might be independent of the clathrin-mediated endocytosis pathway under higher SA conditions (Figure 5A,B). A previous study suggested that the danger-associated peptide pep1 could bind with the membrane-localized leucine-rich repeat receptor kinase PEPR1 and mediate the trafficking of PEPR1 to the vacuole independent of TGN/EE [50]. The pep1 peptide also induced the endocytosis of PIN2 [51]. Therefore, it is very possible that pep1 induced the endocytosis of PIN2 independent of the clathrin-mediated endocytosis pathway.

Sphingolipids (SLs) are an important class of components on the membranes of eukaryotic cells. Studies on human skin fibroblasts have shown that two glycosphingolipid (GSL) analogues were selectively internalized via a clathrin-independent pathway, whereas another SL, sphingomyelin (SM), was internalized by both clathrin-dependent and -independent mechanisms [52]. In myocytes and adipocytes, insulin increased glucose transporter 4 (GLUT4) exocytosis through both clathrin-dependent and -independent endocytosis pathways [53]. We found that PIN2 entered PVC in a clathrin-independent manner and then entered the vesicle for degradation. Combined with a previous report that PIN2 was endocytosed in a clathrin-dependent manner, our findings demonstrated that PIN2 was endocytosed in both clathrin-dependent and non-dependent endocytosis [21]. These results all suggest that a class of substances or proteins can be endocytosed in both clathrin-dependent and -independent manners.

Taken together, a high concentration of SA treatments not only inhibited root growth, root hair development upon gravity stimulus and *PIN2* transcription in the nucleus but also modulated the vacuolar degradation of PIN2, possibly independent of clathrin-mediated endocytic protein trafficking. Under low SA conditions, PIN2 is homogeneously distributed on PM [32]. High SA directly inhibits the transcription of *PIN2* and induces the compartmentalization of membrane nanodomain composed of a lipid-raft structure, which restricts the lateral diffusive movement of PIN2. SA also binds to PP2A, which in turn, affects the phosphorylation of PIN2. The constrained lateral movement of PIN2 causes its clustering and suppressed endocytosis, independent of the clathrin-mediated membrane trafficking pathway during gravitropism. SA can also cross-talk with ROS, which affects the actin dynamics to modulate ARF-GEF-dependent trafficking of PIN2. Finally, PIN2 is sorted from the plasma membrane to vacuolar independent of the clathrin-mediated membrane trafficking pathway (Figure 6). The new mechanism of SA affected gravitropic root growth and development via the regulation of PIN2 protein endocytosis provides a perspective potential for the improvement of economic efficiency of economically important parasitic crop plants [54].

## 4. Materials and Methods

### 4.1. Plant Materials

*Arabidopsis thaliana* ecotype Col-0, the transgenic marker lines DR5-GFP [35], DII-VENUS [36], ProPIN2:*PIN2-GFP* [37], CLC2-mCherry [38], Rha1-mCherry [39], and the generated PIN2-GFP×CLC2-mCherry and PIN2-GFP×Rha1-mCherry cross lines were used.

### 4.2. Growth Condition

*Arabidopsis* seeds were surface-sterilized and sown on ½MS medium containing 1% (*w/v*) sucrose and 1.1% phytoblend (Caisson Labs, PTP01) agar (*w/v*). After vernalization for 2 days at 4 °C, seeds were germinated at 22 °C under a 16:8 h light:dark period with a light intensity of 150 µmol s^−1^m^−2^. Five-day-old seedlings were used for experiments, except for those with specific indications.

### 4.3. Root Gravitropism Assay

Seedlings were vertically cultured for 3 days or 5 days on ½MS medium, and changes in the angle of the root tip beyond the direction of vertical growth was measured using the imageJ software.

### 4.4. Confocal Microscopy Analysis

Images were taken using an LSM-710 confocal microscope (Zeiss) equipped with an argon/krypton laser. The excitation wavelengths for the GFP and mCherry signals were 488 and 587 nm, respectively. For the quantitative fluorescence intensity, the confocal pictures were acquired using strictly identical acquisition parameters (laser power, photomultiplier, offset, zoom factor, and resolution) among each experimental seedling. For co-localization analysis, the Co-localization Finder plugin of ImageJ was used.

### 4.5. Quantitative Real-Time PCR (qRT-PCR)

Total RNA was isolated from the Col-0 seedlings using Trizol reagent (Invitrogen). Complementary DNA was synthesized using the M-MLV Reverse Transcriptase (Promega). qRT-PCR was conducted using the SYBR Green I Master kit (Cham Q Universal SYBR qPCR Master Mix, Vazyme) on a CFX Connect Real-Time System (Bio-Rad). All individual reactions were carried out in triplicate. *ACTIN2* (AT3G18780) was used as an internal control. The primers used for qRT-PCR are listed in Appendix A.

### 4.6. Statistical Analysis

All data were analyzed with Origin 8 and SPSS19. Two-tailed Student’s *t*-test was used for statistical analysis.

## 5. Conclusions

The gravitropic root growth and root hair development of *Arabidopsis* seedlings under high SA stress conditions were investigated. Exogenous application of a high concentration of SA inhibited the gravitropic root growth and root hair development in a dose-dependent manner. The inhibition resulted from both the down-regulated *PIN2* transcription and up-regulated PIN2 endocytosis. Further protein localization analyses in transgenic marker lines indicated that the vacuolar PIN2 degradation might be independent of clathrin-mediated endocytic protein trafficking. Our findings in this study reveal a new underlying mechanism of SA-affected gravitropic root growth and root hair development regulated by the endocytic trafficking PIN2 protein.

## Figures and Tables

**Figure 1 ijms-23-09379-f001:**
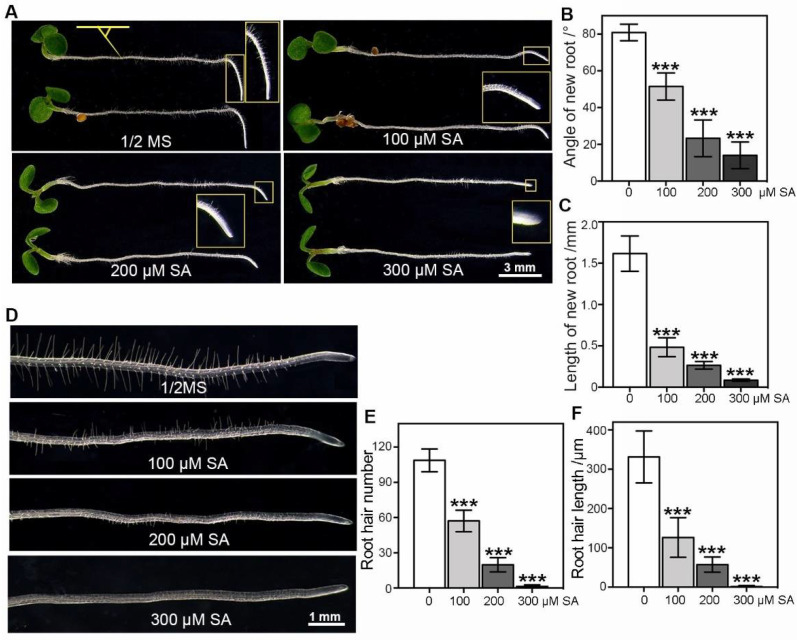
Effects of high-concentration SA on root gravitropic responses and root hair development in *Arabidopsis*. Five-day-old wild-type *Arabidopsis* (Col-0) seedlings grown on ½MS medium were used. (**A**–**C**) Effects of SA treatments on the deviated root tip angles and new root growth. Five-day-old seedlings were transferred to new ½MS medium plates supplemented with 100, 200 or 300 μM SA, and cultured for another 16 h after being orientated at 90°. The inset images show 1.5, 2.0, 3.0, and 4.5× enlarged new root growth, respectively. Scale bar = 3 mm. (**D**–**F**) Effects of SA treatments on root hair development. Five-day-old seedlings transferred to new ½MS medium plates supplemented with 100, 200 or 300 μM SA were vertically cultured for another 48 h. Root hair number and length were measured (n = 6). Scale bar = 1 mm. Student’s *t*-test was performed between the groups (mock vs. SA). Error bar = standard deviation (SD). *p*-values were determined with two-tailed Student’s *t*-test assuming equal variances (*** *p* < 0.001).

**Figure 2 ijms-23-09379-f002:**
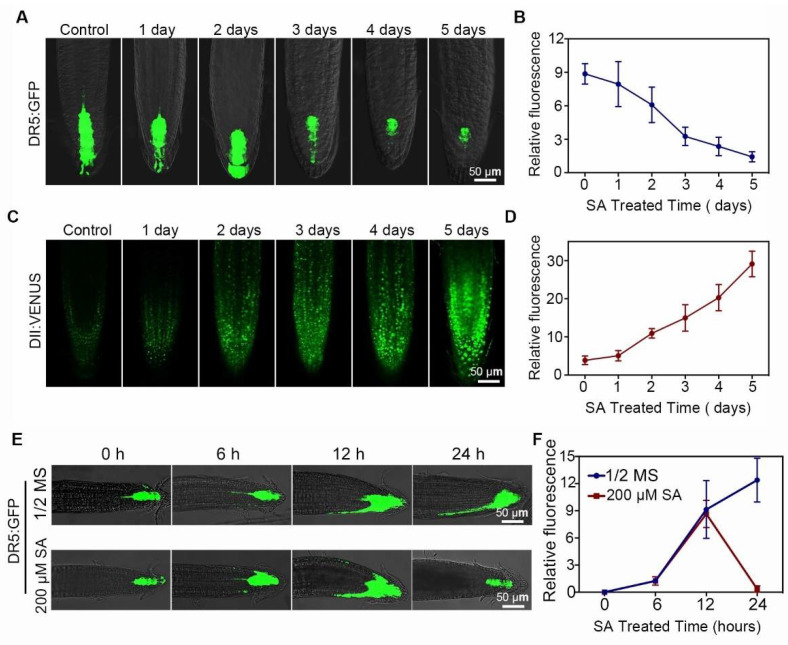
Effects of high-concentration SA on auxin accumulation and transportation. (**A**,**B**) Auxin accumulation in transgenic DR5-GFP seedlings. Three-day-old DR5-GFP seedlings grown on ½MS were treated with 200 μM SA for 1, 2, 3, 4 and 5 days, respectively. DR5-GFP signal (fluorescence intensity) was examined after the respective treatment (n = 4). (**C**,**D**) Auxin degradation in transgenic DII-VENUS seedlings. Three-day-old DII-VENUS seedlings grown on ½MS were treated with 200 μM SA for 1, 2, 3, 4 and 5 days, respectively. DII-VENUS signal (fluorescence intensity) was examined after the respective treatment (n = 4). (**E**,**F**) Auxin distribution in transgenic DR5-GFP seedlings. Five-day-old DR5-GFP transgenic seedlings were transferred to ½MS medium containing 0 or 200 μM SA and cultured for 0 h, 6 h, 12 h and 24 h after 90° re-orientation. DR5-GFP signal (fluorescence intensity) was examined after the respective treatment (n = 4). Error bar = standard deviation (SD).

**Figure 3 ijms-23-09379-f003:**
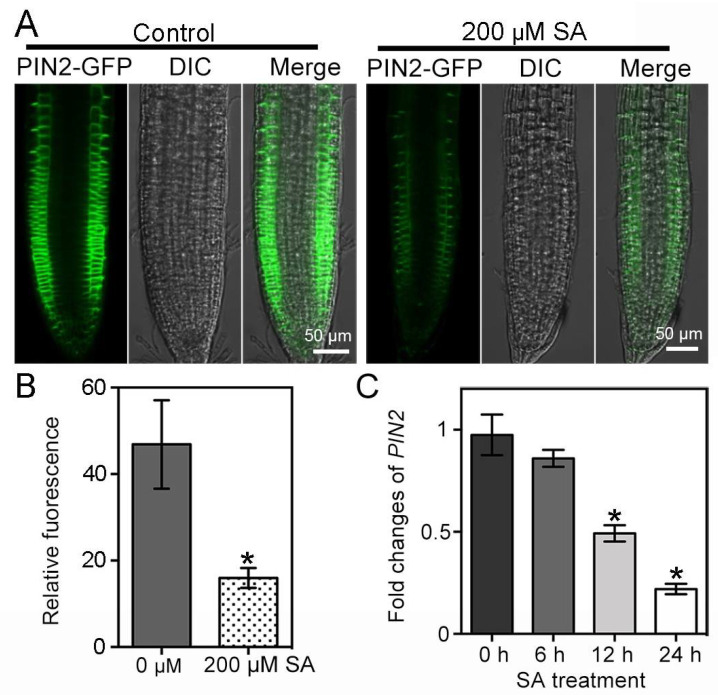
Effects of high-concentration SA on PIN2 accumulation and *PIN2* expression. (**A**,**B**) PIN2 accumulation in the roots of transgenic ProPIN2:*PIN2-GFP* seedlings. Five-day-old ProPIN2:*PIN2-GFP* seedlings were treated with 200 μM SA for 2 days. PIN2-GFP signal was measured in the roots (n = 4). (**C**) qRT-PCR analysis of *PIN2* expression in wild-type *Arabidopsis* (Col-0) seedlings. Sever-day-old seedlings were transferred to 200 μM SA-containing ½MS medium and cultured for 0 h, 6 h, 12 h and 24 h, respectively. Scale bar = 50 μm. Error bar = standard deviation (SD). *p*-values were determined with two-tailed Student’s *t*-test assuming equal variances (* *p* < 0.05).

**Figure 4 ijms-23-09379-f004:**
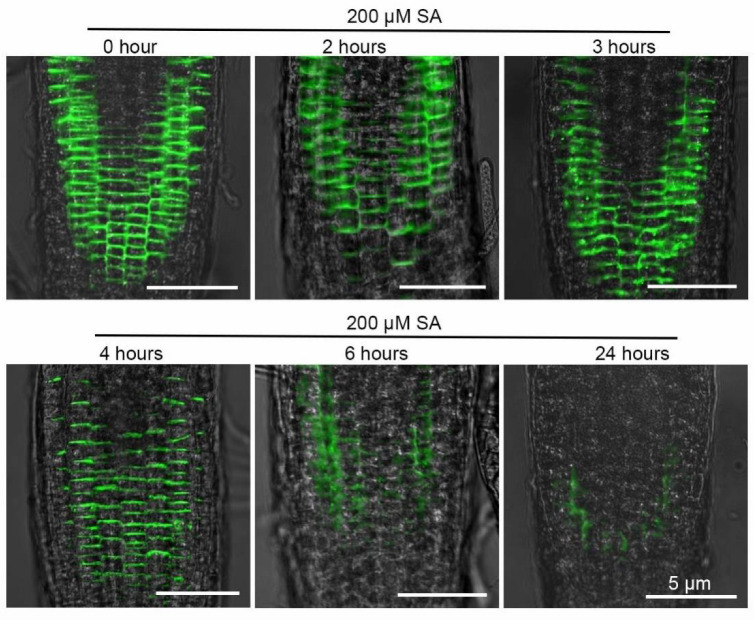
High-concentration SA induces PIN2-GFP degradation. PIN2-GFP fluorescence signal in the roots of five-day-old ProPIN2:*PIN2-GFP* transgenic seedlings treated with 200 μM SA for 0, 2, 3, 4, 6 and 24 h was observed. Scale bar = 5 μm.

**Figure 5 ijms-23-09379-f005:**
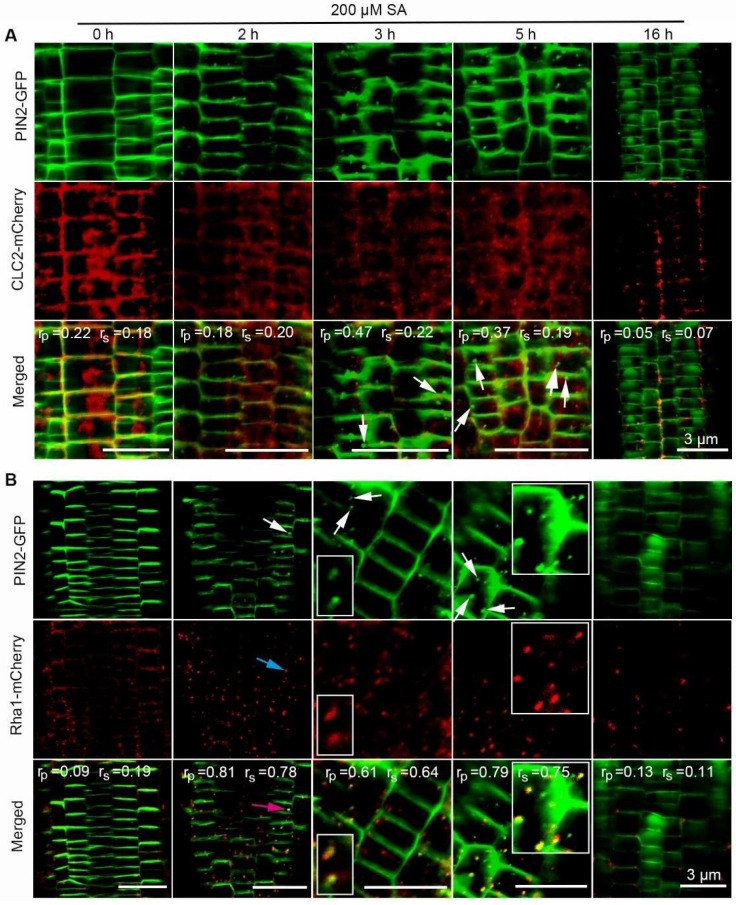
PIN2 is transported to PVC for degradation under a high concentration of SA treatment. Five-day-old transgenic *Arabidopsis* seedlings harboring both PIN2-GFP and CLC2-mCherry (**A**) or PIN2-GFP and Rha1-mCherry (**B**) were treated with 200 μM SA for 2, 3, 5 and 16 h, respectively. The green, red and yellow fluorescence signals respectively represented GFP, mCherry and the co-localized GFP and mCherry proteins. The white, indigo and red arrows respectively denoted the aggregated PIN2-GFP, Rha1-mcherry, and the co-localized PIN2-GFP and Rha1-mCherry proteins. The small and big white boxes respectively showed the 2× and 8× images of the samples treated for 5 and 3 h. Scale bar = 3 μm. Values of Rp and Rs coefficients were calculated.

**Figure 6 ijms-23-09379-f006:**
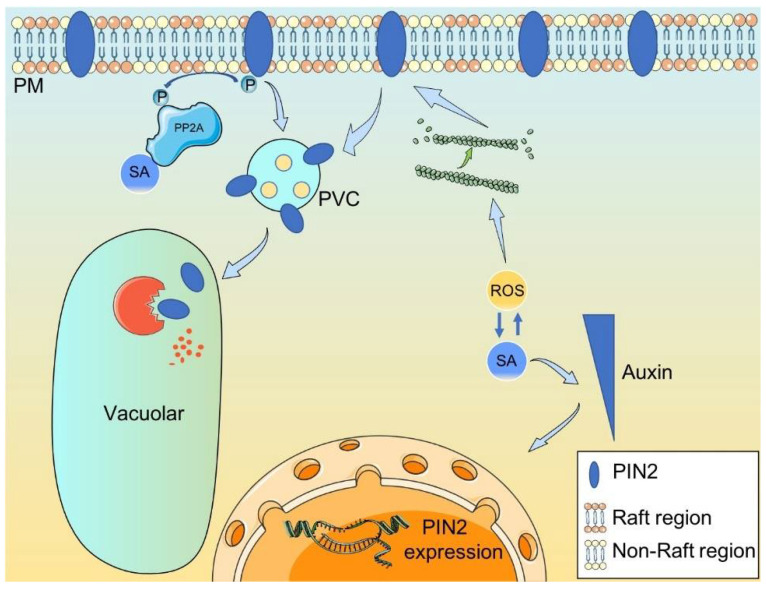
A speculative model of SA regulated auxin transport. High SA directly inhibits the transcription of *PIN2* and induces the compartmentalization of membrane nanodomain, which restricts the lateral diffusive movement of PIN2. SA also binds to PP2A, which in turn affects the phosphorylation of PIN2. The constrained lateral movement of PIN2 causes its clustering and suppressed endocytosis independent of the clathrin-mediated membrane trafficking pathway. SA also cross-talks with ROS, which affects the actin dynamics to modulate ARF-GEF-dependent trafficking of PIN2. Finally, PIN2 is sorted from the plasma membrane to the vacuoles.

## Data Availability

The data presented in this study are available on request from the corresponding author.

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
