# Peer review of "Salicylic Acid Regulates Root Gravitropic Growth via Clathrin-Independent Endocytic Trafficking of PIN2 Auxin Transporter in Arabidopsis thaliana"

_ijms, 2022, doi:10.3390/ijms23169379_

Round 1

Reviewer 1 Report

Journal             IJMS (ISSN 1422-0067

Manuscript ID IJMS -1856340

Type     Article

Title

Salicylic acid regulates root gravitropic growth via clath-rin-independent endocytic trafficking of PIN2 auxin transporter in Arabidopsis thalianaAuthors

Houjun Zhou , Haiman Ge , Jiahong Chen , Xueqin Li , Hongxia Zhang * , Lei Yang , Yuan Wang *

*Section Molecular Plant Sciences

 Salicylic acid regulates root gravitropic growth via clath-2 rin-independent endocytic trafficking of PIN2 auxin transporter 3 in Arabidopsis thaliana

Q: “The topic is very interesting. It is likely to have a great impact on the advancement of this field.

It will be reasonable to present a small comparison between clathrin-dependent and clathirin- independent routes in terms of the enzymes and/or genes involved in the two routes.

 The phytohormone salicylic acid (SA) plays a crucial role in plant growth and devel-14 opment.

Q: Is there any role of the stringolactones in this pathway?

  Please see if the following published paper proves helpful in strengthening the scientific basis of the present document

             REF:   https://doi.org/10.3390/microorganisms9040774

Q: Our findings pro-26 posed a new underlying mechanism of SA affected gravitropic root growth and root hair devel-27 opment via the regulation of PIN2 gene transcription and PIN2 protein endocytosis in plants.

      Which potential gains are expected through this alternate route: Energy efficiency or Economic efficiency?

      Which other crop plants are likely to benefit from this mechanism?

              Information from the following documents will prove helpful in enhancing the perspective of this study:

                https://doi.org/10.3390/microorganisms9040774

                https://doi.org/10.1007/s12088-021-00979-7

                https://doi.org/10.1016/j.biortech.2019.122346

Author Response

We greatly appreciate the efforts of you for the constructive comments that have helped shape this manuscript into a better form. We have addressed all the concerns by either editing the manuscript or clarifying the details. Please see the detailed point-by-point response added below.

Sincerely,

Hongxia Zhang

Point by point response to Reviewer 1

Specific comments

1, Salicylic acid regulates root gravitropic growth via clathrin independent endocytic trafficking of PIN2 auxin transporter in Arabidopsis thaliana. The topic is very interesting. It is likely to have a great impact on the advancement of this field.

It will be reasonable to present a small comparison between clathrin-dependent and clathirin-independent routes in terms of the enzymes and/or genes involved in the two routes.

Response: Thank you for your comments. We have added a small comparison between clathrin-dependent and clathirin-independent routes in the ‘Discussion’ section as described below: Sphingolipids (SLs) are an important class of components on the membranes of eukaryotic cells. Studies on human skin fibroblasts have shown that two glycosphingolipid (GSL) analogues were selectively internalized via a clathrin-independent pathway, whereas another SL, sphingomyelin (SM), was internalized by both clathrin-dependent and -independent mechanisms (Puri et al., JCB, 2001. doi: 10.1083/jcb.200102084). In myocytes and adipocytes, insulin increased glucose transporter 4 (GLUT4) exocytosis through both clathrin-dependent and independent endocytosis pathways (Antonescu et al., Traffic, 2008. doi/10.1111/j.1600-0854.2008.00755.x). We found that PIN2 entered PVC in a clathrin-independent manner, and then entered the vesicle for degradation. Combined with previous report that PIN2 was endocytosed in a clathrin-dependent manner, our findings demonstrated that PIN2 was endocytosed in both clathrin-dependent and non-dependent endocytosis (Du et al., PNAS, 2013. doi:10.1073/pnas.1220205110). These results all suggest that a class of substances or proteins can be endocytosed in both clathrin-dependent and -independent manners.

2, The phytohormone salicylic acid (SA) plays a crucial role in plant growth and development.

Is there any role of the stringolactones in this pathway? Please see if the following published paper proves helpful in strengthening the scientific basis of the present document

REF: https://doi.org/10.3390/microorganisms9040774.

Response: Thank you. Yes, similarly, strigolactones (SLs) interact with nitric oxide to regulate root system architecture by altering auxin signaling and transport in Arabidopsis thaliana. They also function as signal molecules for the communication between plants and bacteria, and play a significant role in drought resistance. In addition, SLs interfere with the effects of auxin on PIN2 targeting, trafficking and clathrin-mediated endocytosis (Zhang et al., Nature Commun. 2020. doi:10.1038/s41467-020-17252-y). We have added the relevant information in the ‘Discussion’ section of the revised manuscript.

3, Our findings proposed a new underlying mechanism of SA affected gravitropic root growth and root hair development via the regulation of PIN2 gene transcription and PIN2 protein endocytosis in plants.

Which potential gains are expected through this alternate route: Energy efficiency or Economic efficiency?

Which other crop plants are likely to benefit from this mechanism?

Information from the following documents will prove helpful in enhancing the perspective of this study:

https://doi.org/10.3390/microorganisms9040774

https://doi.org/10.1007/s12088-021-00979-7

https://doi.org/10.1016/j.biortech.2019.122346.

Response: Thank you. Through this alternate route, economic efficiency potential gain can be expected. Economically important parasitic plants, and crops with symbiotic rhizosphere microorganisms such as soybean, peanut, blueberry will likely to benefit from this mechanism. We have added the relevant information in the last paragraph of ‘Discussion’ section of the revised manuscript.

Reviewer 2 Report

Comments to the authors

The article entitled “Salicylic acid regulates root gravitropic growth via clathrin-independent endocytic trafficking of PIN2 auxin transporter in Arabidopsis thaliana” is nicely presented. The study showed the effect of gravitropism on root development and growth. Although the manuscript is organized well but still some points need to be discussed in details. Below are comments for authors to improve manuscript before acceptance.

1.      In the abstract the sentence at lines 20-26 is long and difficult to read. It is highly recommended to rephrase the lines to clarify the meaning.

2.      In figure 1 provide an image that clearly shows the difference in root length at different conc. of SA

3.      Provide scale bar for each part of the figure separately on the figures rather than describe in the figure legends.

4.      In the discussion section authors should explain “under which circumstances SA induces clathrin independent-mediated PIN2 degradation and when stimulates PIN2 hyperclustering during gravitropism as described in “Salicylic acid regulates PIN2 auxin transporter hyperclustering and root gravitropic growth via Remorin‐dependent lipid nanodomain organisation in Arabidopsis thaliana. New Phytol. 2021, 229, 963-978”

5.      ABA also controls PIN2 and inhibit auxin transport, refer to the article “Abscisic Acid Regulates the Root Growth Trajectory by Reducing Auxin Transporter PIN2 Protein Levels in Arabidopsis thaliana. Front. Plant Sci., 04 March 2021”. Add discussion about how ABA and SA control auxin transport differently?

6.      Article contains redundancy. Same sentences are repeated at several places in the text. Rephrasing of sentences is required to minimize redundancy.

7.      Italicize the word ‘Arabidopsis’ throughout the whole manuscript.

8.      There are several grammatical errors and typos throughout the manuscript. Therefore, a careful proofread is required at the revision stage.

Author Response

We greatly appreciate the efforts of you for the constructive comments that have helped shape this manuscript into a better form. We have addressed all the concerns by either editing the manuscript or clarifying the details. Please see the detailed point-by-point response added below.

Sincerely,

Hongxia Zhang

Point by point response to Reviewer 2

General comments

The article entitled “Salicylic acid regulates root gravitropic growth via clathrin-independent endocytic trafficking of PIN2 auxin transporter in Arabidopsis thaliana” is nicely presented. The study showed the effect of gravitropism on root development and growth. Although the manuscript is organized well but still some points need to be discussed in details. Below are comments for authors to improve manuscript before acceptance.

Response: Thank you for your comments.

Specific comments

  1. In the abstract the sentence at lines 20-26 is long and difficult to read. It is highly recommended to rephrase the lines to clarify the meaning.

Response: Thank you. We have rephrased the lines as suggested.

  1. In figure 1 provide an image that clearly shows the difference in root length at different conc. of SA.

Response: Thank you. We have provided enlarged pictures to clearly show the difference in root length at different conc. of SA in Figure 1A.

  1. Provide scale bar for each part of the figure separately on the figures rather than describe in the figure legends.

Response: Thank you. We have provided scale bar for each part of the figure separately on the figures following your suggestion.

  1. In the discussion section authors should explain “under which circumstances SA induces clathrin independent-mediated PIN2 degradation and when stimulates PIN2 hyperclustering during gravitropism as described in “Salicylic acid regulates PIN2 auxin transporter hyperclustering and root gravitropic growth via Remorin-dependent lipid nanodomain organisation in Arabidopsis thaliana. New Phytol. 2021, 229, 963-978”.

Response: Thank you. Ke et al. reported that SA stimulated PIN2 hyperclustering during gravitropism at a concentration of 100 μM (New Phytol. 2021, 229, 963-978). We treated Arabidopsis seedlings with 200 μM SA and found that SA induced clathrin independent-mediated PIN2 degradation, suggesting different effects of different SA dosages on PIN2. We have added the relative information in the revised manuscript as described by Ke et al.

  1. ABA also controls PIN2 and inhibits auxin transport, refer to the article “Abscisic Acid Regulates the Root Growth Trajectory by Reducing Auxin Transporter PIN2 Protein Levels in Arabidopsis thaliana. Front. Plant Sci., 04 March 2021”. Add discussion about how ABA and SA control auxin transport differently?

Response: Thank you. We have discussed how ABA and SA control auxin transport differently in the ‘Discussion’ section of the revised manuscript.

  1. Article contains redundancy. Same sentences are repeated at several places in the text. Rephrasing of sentences is required to minimize redundancy.

Response: Thanks. We have rephrased the redundant sentences as suggested.

  1. Italicize the word ‘Arabidopsis’ throughout the whole manuscript.

Response: Thank you. We have italicized the word ‘Arabidopsis’ throughout the whole manuscript.

  1. There are several grammatical errors and typos throughout the manuscript. Therefore, a careful proofread is required at the revision stage.

Response: Thank you. We have performed a careful proofread.

Round 2

Reviewer 1 Report

Acceptable